# Assessing the Impact of Use and Trust in Different Sources of Information on COVID-19 Vaccination Uptake in Saudi Arabia (SA) Using the COVID-19 Vaccine Hesitancy and Resistance in SA (CoV-HERSA) Tool

**DOI:** 10.3390/tropicalmed7110375

**Published:** 2022-11-14

**Authors:** Anwar A. Sayed

**Affiliations:** 1Department of Medical Microbiology and Immunology, Taibah University, Medina 42353, Saudi Arabia; anwar.sayed13@imperial.ac.uk; 2Department of Surgery and Cancer, Imperial College London, London SW7 2AZ, UK

**Keywords:** COVID-19, public health, Saudi Arabia, social media, vaccine hesitancy

## Abstract

COVID-19 vaccination has been the cornerstone measure to tackle the severe morbidity and mortality of the ongoing global pandemic. However, vaccine hesitancy and resistance were observed in different populations, including Saudi Arabia (SA), yet such hesitancy was not accurately measured, nor were its influencing factors determined. The COVID-19 Vaccine Hesitance and Resistance in SA (CoV-HERSA) tool was developed, validated, and distributed to 387 participants to accurately measure their COVID-19 vaccine hesitancy and its influencing factors. Different chronic conditions affected participants’ CoV-HERSA differently, with those having autoimmune disorders having the highest CoV-HERSA scores. Previous exposure to COVID-19 significantly increased the CoV-HERSA scores. The use and trust of the different sources of information had a significant influence on the CoV-HERSA scores. Those who used newspapers and healthcare practitioners as their source of information had the highest CoV-HERSA scores, while those who relied on family/friends had the lowest scores. The CoV-HERSA is a validated tool that accurately reflects participants’ willingness and hesitancy to COVID-19 vaccination and can be used to explore the influence of different factors on the participants’ attitudes towards the COVID-19 vaccine.

## 1. Introduction

The novel coronavirus disease (COVID-19) pandemic has posed a significant challenge to the medical field. Its complex pathophysiology, affecting different body systems and involving various immune cells, and its unpredicted course made it a medical dilemma. Despite the advances in detecting and predicting the infection course [1,2], no standard line of treatment has been established globally and hopes relied on extensive efforts toward developing an effective vaccine.

Pfizer/BioNTech (PB) and AstraZeneca (AZ) COVID-19 vaccines were among the earliest to develop and FDA-approved in the United States and the United Kingdom. Early vaccine studies showed that PB and AZ efficacy exceeded 70% protection in the United States and the United Kingdom, respectively [3,4]. 

Since the beginning of the COVID-19 pandemic, Saudi Arabia (SA) has been one of the earliest to take active and progressive measures to protect its people [5]. As part of its ongoing efforts, SA was also one of the earliest countries to approve both PB and AZ vaccines. Once approved and imported, SA has started a national mass COVID-19 vaccination program with over 6 million doses being administered. This figure indicates that less than one-tenth of SA’s population received double doses of the vaccine, which was not expected given the vaccine anticipation at the beginning of the pandemic. As the number of confirmed COVID-19 cases has increased again since March 2022, vaccine uptake has become an urgent public health necessity that needs to be encouraged. This is further mandated as a single dose of COVID-19 vaccinations was not enough to prevent its spread [6].

Previous studies demonstrated that COVID-19 vaccine uptake is influenced by various factors, including anxiety about the infection [7], worries about the unforeseen future, and mistrust in the information regarding vaccine safety and efficacy [8]. This antivaccination attitude contributed to the exposure of false information through alternative and unchecked sources such as social media [9].

The pandemic has been associated with a wave of misinformation, which was easily accessible to the public. This study aims to identify the misconceptions regarding the COVID-19 vaccines and how previous personal experiences with COVID-19, such as previous infection or death of a relative, might influence the vaccine perception. It also aims to assess the sources of these misconceptions and how they influence the public’s opinion, contributing to the COVID-19 vaccine hesitancy and resistance in the Saudi population.

This study will attempt to assess the public knowledge and perceptions of the COVID-19 vaccine and its misinformation. Identifying the significant misconceptions around the COVID-19 vaccine and its sources will allow public health practitioners to develop educational programs that address these misconceptions. It will also better inform healthcare authorities on which media is best to address them.

## 2. Materials and Methods

### 2.1. Methodological Approach

This descriptive cross-sectional (observational) study attempted to assess participants’ hesitance and resistance to COVID-19 vaccination. This was done by developing and validating a COVID-19 Vaccine Hesitancy and Resistance in Saudi Arabia (CoV-HERSA) instrument. The CoV-HERSA instrument is made of ten statements, five that are either positive or negative towards COVID-19 vaccination. Each statement is rated on a five-point scale from −2 to +2 based on the participants’ responses. Participants’ responses indicated their agreement with the provided statements and scored accordingly. Neutral answers, ‘I do not know’, are given a score of 0. The statements and their scoring are detailed (Table A1).

The survey was distributed via social media platforms, such as Facebook, Twitter, and WhatsApp, in keeping with the guidance on the medical use of social media [10]. 

### 2.2. Piloting the Questionnaire

The questionnaire was piloted with 30 participants in-person before it was sent out to the participants. The pilot was to assess the questionnaire’s clarity and ease of answering and address any potential issues, such as questions that are difficult to read, spelling issues, or better wording. Test-retest and inter-rater reliability were determined with a Cronbach’s alpha score of 0.8.

### 2.3. Data Management

The questionnaire results of each participant were stored in a password-protected folder. Multiple copies of this folder data were stored on a secure cloud-based server that requires a username, password, and physical hard drives. Only the principal researcher had access to the raw data for this project’s duration. Data were managed following Taibah University Research Data Management Policy.

### 2.4. Statistical Analysis

The collected survey data were analyzed using descriptive and analytical quantitative statistical methods using GraphPad Prism version 9.3 (GraphPad Software, San Diego, CA, USA). 

The distribution of the numerical variables (Age and CoV-HERSA score) was determined using Shapiro -Wilk Test. Parametric and nonparametric methods were used on datasets with a normal (gaussian) and not a normal distribution, respectively. Unpaired student t-test and Mann-Whitney U tests were used to compare 2 parametric and nonparametric datasets, respectively. To compare more than two groups of data, one-way analysis of variant (ANOVA) with Bonferroni correction for multiple comparisons was used for parametric datasets. The Kruskal-Wallis test with Dunn’s correction for multiple comparisons was used for nonparametric datasets. 

The interaction between the use, trust, and the CoV-HERSA scores were evaluated using Two-way ANOVA, followed by Tukey correction for multiple comparisons. The interaction between the use and trust in the different sources of information and their impact on the participants’ CoV-HERSA scores was assessed using Three-way ANOVA. Statistical significance was denoted at a *p*-value less than 0.05.

## 3. Results

### 3.1. Participants’ Characteristics

A total of 387 participants responded to the survey, of which 193 were male (49.9%), and their median age was 34. 83.7% of the participants were smokers (*n* = 324), with the majority of them (76.4%) having gone through high education or higher (Bachelor’s and Postgraduate degrees). 

Over half of the participants (51.7%) resided in Madinah, while the remaining were from other administrative provinces such as Makkah, Riyadh, and the Eastern Province of Saudi Arabia. Almost half of the participants are part of an educational establishment, either staff or students. The detailed characteristics of the study participants are described (Table 1).

### 3.2. Personal Medical Experience with COVID-19

Participants’ previous exposure to medical conditions, either personally or through a relative/friend, could influence their attitude toward COVID-19 vaccination. Similarly, pregnancy and previous immunization with different vaccines, e.g., the Hepatitis B vaccine, could shape their response to COVID-19 vaccination. Hence, this was evaluated as part of this study.

The majority (71.1%) of the participants of this study did not complain of any chronic medical conditions. However, over 70% of the study cohort had an immediate family member with Diabetes Mellitus, either type 1 or 2. Only six female participants were pregnant (3.1%), while 130 participants (33.6%) had a pregnant immediate family member.

Most of the study participants (63%) received non-COVID-19 vaccinations such as Hepatitis B or seasonal influenza. A detailed breakdown of the participants’ previous medical experience is described (Table 2).

Forty-two participants were diagnosed with COVID-19 through a PCR test, and only two were admitted to a hospital. On the other hand, 345 participants had an immediate family/friend who was diagnosed with COVID-19, with over 30% of those admitted to a hospital (*n* = 91) or died (*n* = 120) from COVID-19. A detailed breakdown of the participants’ experience with COVID-19 is summarized (Table 3).

### 3.3. The Impact of Participants’ Characteristics and Experience with COVID-19 on Their CoV-HERSA Scores

The participants’ median CoV-HERSA score was 5 (IQR 2–9). Male participants scored higher than female participants. However, such a difference was not statistically significant. There were differences in the CoV-HERSA score between participants based on their education level. Those with intermediate school certificates had the lowest score, whereas those with postgraduate certificates had the highest median scores, −1 and 6, respectively. The workplace’s nature did not significantly influence the CoV-HERSA score (*p*-value > 0.05), as participants from different work environments had comparable scores. 

There was a significant difference between participants based on their region of residence (*p*-value < 0.01). Those from the Eastern province had the highest CoV-HERSA score (median of 11), whereas those from the Northern Borders had the lowest score (median of −5) (Figure 1). 

Interestingly, the type of chronic condition that participants had significantly influenced their CoV-HERSA score (*p*-value < 0.05). Those with autoimmune disorders had the highest CoV-HERSA score (median of 16), whereas those with Type 1 Diabetes Mellitus and Immunodeficiency were the lowest-scoring participants, with scores of −2.5 and −2, respectively (Figure 2). Healthy participants scored higher than those with chronic conditions. However, such a difference was not statistically significant. 

Regarding the impact of previous experience with COVID-19 on the participants’ vaccine hesitancy and resistance, those with previous COVID-19 infection had a higher CoV-HERSA score, although not statistically significant (Figure 3A). Similarly, those with family and friends who were hospitalized due to COVID-19 had a slightly higher CoV-HERSA score (*p*-value > 0.05) (Figure 3B). Interestingly, those who had an immediate relative infected with COVID-19 had a significantly lower score compared to those who did not have any immediate family with a previous COVID-19 infection (4 vs. 5, *p*-value < 0.05) (Figure 3C). No significant difference was observed between those with a relative who died from COVID-19 and those who did not (Figure 3D).

### 3.4. The Impact of the Use and Trust in Different Sources of Information on Participants’ CoV-HERSA Scores

The studied participants were asked about their use of different information sources, namely, social media platforms, TV/radio, newspapers/magazines, family/friends, and healthcare practitioners. Their CoV-HERSA scores were compared based on their answers about the use or lack of these sources.

Based on the Two-way ANOVA testing, there was a significant interaction between the different sources of information and their use on the CoV-HERSA scores (*p*-value < 0.05). Those who used newspapers/magazines had the highest median CoV-HERSA score of 9, followed by those who used healthcare practitioners with a median score of 7. On the other hand, those who did not use family/friends as their source of information had the highest median score of 9, followed by those who did not use social media with a median score of 7. 

The most significant differences observed between the use and the lack of a particular source are family/friends (*p*-value < 0.01) and healthcare practitioners (*p*-value < 0.05). Interestingly, it seemed that the use, or lack of, TV/radio and social media did not significantly affect participants’ CoV-HERSA scores (Figure 4). 

Similarly, the study participants were asked about their trust, or lack of it, in the different sources of information. Then their COVID-19 vaccine hesitancy or resistance was compared according to their trust in these sources.

Based on the Two-way ANOVA testing, there was a significant interaction between the trust, or lack of it, in the different sources of information on participants’ CoV-HERSA scores (*p*-value < 0.05). Those who trusted TV/radio had the highest median CoV-HERSA score of 7, followed by those who trusted social media with a median score of 6.5. On the other hand, those who did not trust family/friends as their source of information had the highest median score of 8, followed by those who did not trust social media as their source of information with a median score of 6. 

The most significant differences observed between trust, and the lack of it in a particular source, are TV/radio (*p*-value < 0.01) and healthcare practitioners (*p*-value < 0.05). Interestingly, it seemed that the trust or lack of trust in newspapers/magazines and social media did not significantly affect participants’ CoV-HERSA scores (Figure 5).

Lastly, it was tested whether both the use and trust or their lack of different sources of information interacted and affected the CoV-HERSA scores of the participants. The 3-way ANOVA testing showed a significant interaction between the different sources of information and their use and trust (*p*-value < 0.01). Similarly, there were significant differences in the CoV-HERSA scores between those who used and trusted the different sources of information and those who did not (*p*-value < 0.05) (Figure 6).

## 4. Discussion

This study has attempted to measure the hesitancy and resistance of the Saudi population toward the uptake of COVID-19 vaccines. Although several studies have attempted to determine vaccine hesitancy in SA, they have used different questionnaires and based their findings on the participants’ demographic characteristics. However, this is the first to use a validated tool (CoV-HERSA) that can be used across different populations and settings. This study also aimed to demonstrate the impact of the use and trust in the different sources of information on the vaccine’s hesitancy and resistance.

Upon examining the impact of participants’ characteristics on their vaccine hesitancy, there was no significant difference in the CoV-HERSA score between male and female participants. Such finds are consistent with the previous work of Al-Mansour and colleagues [11], as well as Al-Mohaithef’s [12] and Yahia’s [13]. On the other hand, Alfageeh and her team contradicted this finding, which showed that women are more resistant to the COVID-19 vaccine than men [14]. Such difference may be attributed to some of the misconceptions regarding vaccines’ side effects, e.g., negative impact on women’s fertility [15]. 

In this study, no direct or indirect correlation was found between the participants’ age and their CoV-HERSA score, indicating a lack of association between age and the participants’ hesitancy toward the COVID-19 vaccine. This finding aligns with what Qattan and her colleagues described [16]. Interestingly, previous studies had conflicting findings on the effect of age, e.g., older people are more or less resistant to COVID-19 vaccines. These observational differences are likely due to the different baseless age categorizations used in each study, which seem to skew the findings. For example, Al-Mohaithef and Padhi categorized their participants’ age into 18–25, 26–35, 36–45, and above 45 years old [17]. On the other hand, Sallam and colleagues had different age categorizations: 16–21, 22–26, 27–39, 40 years, and older [18]. Hence, the ability to assign a specific value to participants’ hesitancy toward the vaccine using the CoV-HERSA tool allows for more precise age association without the need to categorize participants according to their age.

The impact of previous COVID-19 infection, hospitalization, and death due to COVID-19 was assessed on vaccine hesitancy. Expectedly, those who previously experienced COVID-19, were mildly or admitted to the hospital, or had a relative/friend who died because of COVID-19 had higher CoV-HERSA scores than those who did not, i.e., were more willing to get vaccinated. Similar findings were demonstrated by Al-zahrani and colleagues in their study among medical staff in Saudi Arabia [19]. Interestingly, participants with different chronic conditions had different attitudes towards COVID-19 vaccines, i.e., significant differences in their CoV-HERSA scores. Although not documented previously, such a finding was suggested by Dhama and his colleagues as they stated, “*people with comorbidities can have a great degree of fear of COVID-19 as well as a different attitude to COVID-19 vaccination*” [20].

This study explores an interesting yet commonly unaddressed in the literature. In previous studies, the use, or lack of, social media platforms, such as Twitter or Facebook, were assessed about various aspects of their users. For example, studies evaluated the impact of social media on consumers’ behaviors [21] as facilitators of negative attitudes and behaviors among adolescents [22] or the role of social media on recycling behavior [23]. Most, if not all, of these studies, assume that social media users trust the information presented through these platforms and subsequently base their findings on such assumptions. However, in this study, an attempt was made to differentiate between these two aspects, the use and trust in social media and other sources of information. Indeed, this is clearly demonstrated as the CoV-HERSA scores of using and trusting are neither identical nor correlating.

In this study, an attempt was made to determine the impact of using and trusting different sources of information on the participants’ willingness to get vaccinated. Expectedly, there were significant differences between the different sources of information. For example, those who do not use social media have a higher CoV-HERSA score. This is in line with the work of Temsah and colleagues, who demonstrated that almost 50% of their study population use different forms of social media, which impact their attitudes towards COVID-19 vaccination [18,24]. On the other hand, using and trusting healthcare practitioners as the source of information positively influenced participants’ CoV-HERSA scores, being higher than those who did not. Such finding was validated by the work of Sallam and his team, who have found that trusting social media was accompanied by a belief in conspiracy theories, leading to distrust and resistance toward COVID-19 vaccination [18].

Vaccine acceptance, hesitancy, and resistance are not binary attitudes, i.e., they are not “All or none” attitudes. These attitudes represent a spectrum, from the far positive end, acceptance, to the far negative one, resistance. Hence, the binary approach used by most of the studies conducted in SA to assess the acceptability, hesitancy, and resistance toward COVID-19 vaccination is not fit for purpose. Such spectrum should be measured using a tool that reflects the nature of such attitudes, such as the CoV-HERSA tool. This tool, which scores span from −20 (completely resistant) to 20 (completely accepting), measures participants’ attitudes toward the vaccine more precisely. Similarly, the tool can be easily correlated with participants’ age to assess better the influence of age on their vaccine acceptance or resistance without the need for age categorization. 

This study is one of its kind in SA to assess the impact of use and trust in the different sources of information on the participants’ attitudes towards the COVID-19 vaccines. In the current age of mass information, anyone with an internet connection can access millions of online sources. This medium could easily facilitate the spread of misinformation and biased opinions on vaccines [25]. In a study by Li and colleagues, they found that over 27% of the top videos on Youtube using the search term COVID-19” had false information in them [26]. Hence, learning about where participants get their information and whether they trust these sources is essential to determine how they influence their attitudes toward the vaccine. Hou et al. studied the impact of social media on participants from New York (United States), London (United Kingdom), Mumbai (India), Sao Paulo (Brazil), and Beijing (China). Over 12,000 Twitter users lacked confidence in COVID-19 vaccine safety [27].

This study is not without limitations. As a cross-sectional study, the findings reflect a section of the Saudi population at the time of the study. Such attitudes may have changed from the beginning of the pandemic to the current time. Another limitation of the study is the number of participants enrolled, despite the effort to maximize participation. Compared to some previous studies, the smaller number of participants may not accurately represent the Saudi population. Another limitation is that most of the study participants are between 24–43 years of age, with a relatively high level of education (76.4%) with a Bachelor’s degree. Such factors may skew the results and prevent their generalizability to the Saudi population.

Future studies could be directed to a broader population sector to assess the impact of the different sources of information on their perception of COVID-19 vaccinations.

## 5. Conclusions

COVID-19 vaccination is key to tackling the ongoing global pandemic. A clearer understanding of its hesitancy and resistance among the population is vital to public health policymakers. The presented CoV-HERSA tool in this study can serve as a direct, precise, and sensitive tool to address the common misconceptions and beliefs about COVID-19 vaccinations, which can be reused reliably in different populations. The CoV-HERSA tool could be used in future studies on a nationwide scale to obtain the population’s detailed perception of COVID-19 vaccination, which would better inform health policymakers. 

## Figures and Tables

**Figure 1 tropicalmed-07-00375-f001:**
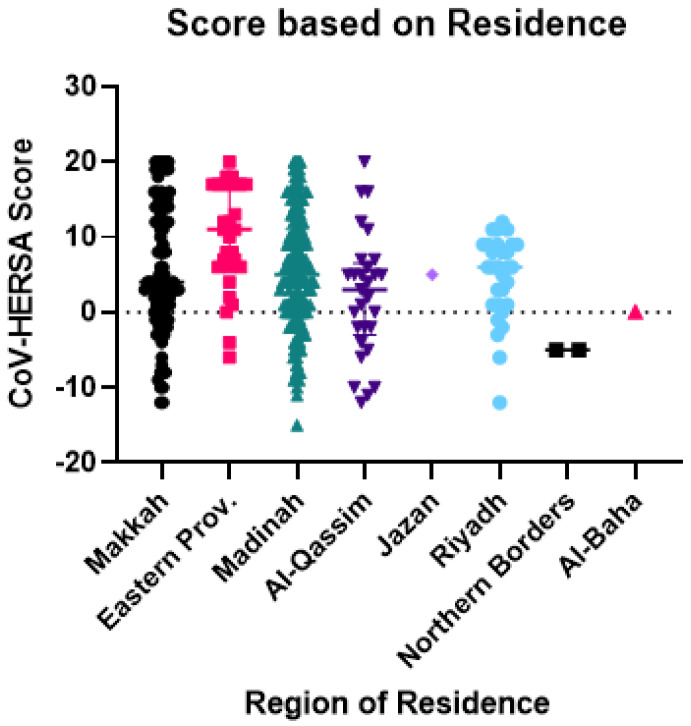
CoV-HERSA score comparison based on the region of residence. The scatterplot demonstrates the CoV-HERSA scores of participants according to their region of residence. These regions are Makkah (Black circles), Eastern Province (Pink squares), Madinah (Green triangles), Al-Qassim (Dark Purple Triangles), Jazan (Purple diamond), Riyadh (Blue circles), Northern Borders (Black squares), and Al-Baha (Red triangle). Lines reflect the median score with an interquartile range where applicable. The dotted line represents a CoV-HERSA score of 0, and the Kruskal-Wallis test with Dunn’s correction was used.

**Figure 2 tropicalmed-07-00375-f002:**
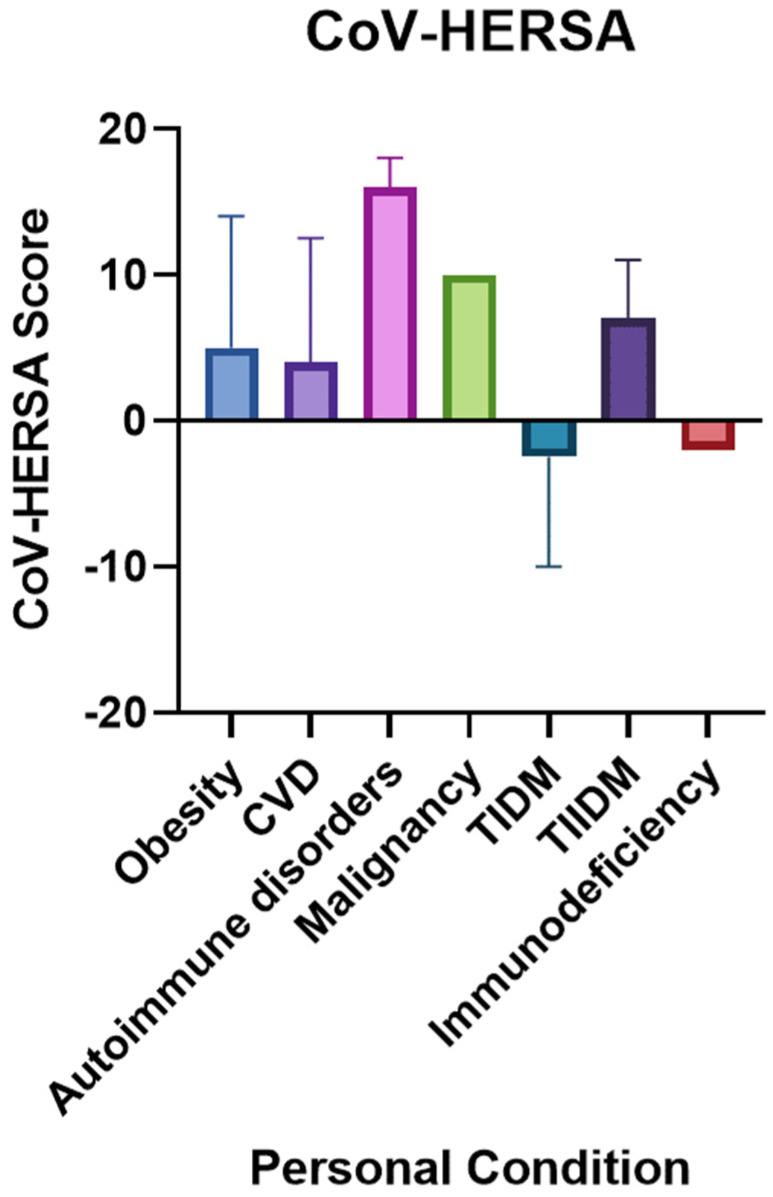
CoV-HERSA scores comparison based on the participants’ comorbidities. The bar chart demonstrates the CoV-HERSA according to the conditions they suffer from, which are obesity (Blue), CVD (Purple), autoimmune disorders (Pink), malignancy (Green), TIDM (Navy blue), TIIDM (Dark Purple) or immunodeficiency (Red). Bar charts demonstrate median values and interquartile ranges where applicable. Kruskal-Wallis test with Dunn’s correction was used. CVD: Cardiovascular diseases; TIDM: Type 1 Diabetes Mellitus; TIIDM: Type 2 Diabetes Mellitus.

**Figure 3 tropicalmed-07-00375-f003:**
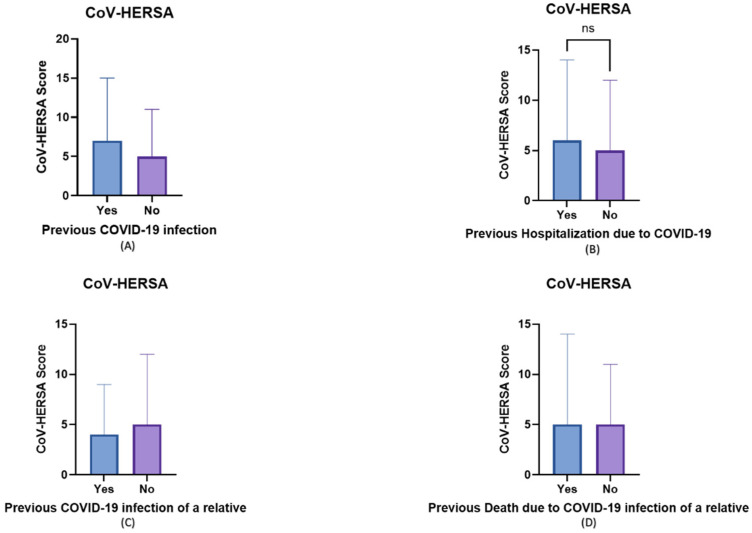
The participants’ experience with COVID-19 impacted their COVID-19 vaccine uptake. Bar charts demonstrate the impact of various aspects of participants’ experience with COVID-19 (Yes/Blue) or lack of (No/Purple) on the participants’ CoV-HERSA scores. These experiences include (**A**) the previous infection, (**B**) the previous hospitalization due to COVID-19, (**C**) the previous infection of a relative, and (**D**) the previous death of a relative due to COVID-19. Bar charts demonstrate median values and interquartile range, and Mann-Whitney U tests were used in these comparisons.

**Figure 4 tropicalmed-07-00375-f004:**
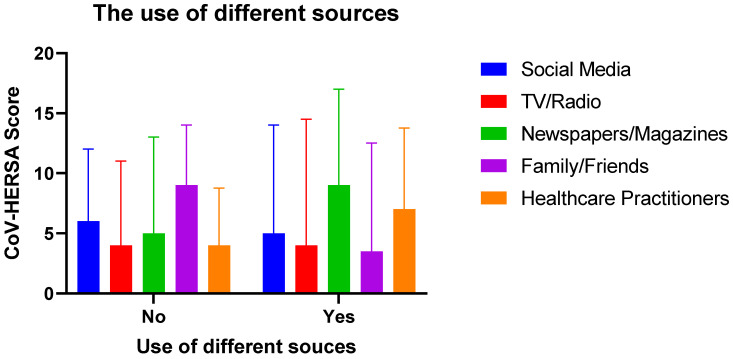
Participants’ CoV-HERSA scores are based on their use of different sources of information. The bar chart demonstrates the different CoV-HERSA scores of participants based on whether or not they use the different sources of information. These sources are social media (Blue), TV/radio (Red), Newspapers/magazines (Green), family/friends (Purple), and Healthcare practitioners (Orange). The bar chart demonstrates median scores with the interquartile range. Two-way ANOVA followed by Tukey correction for multiple comparisons were used.

**Figure 5 tropicalmed-07-00375-f005:**
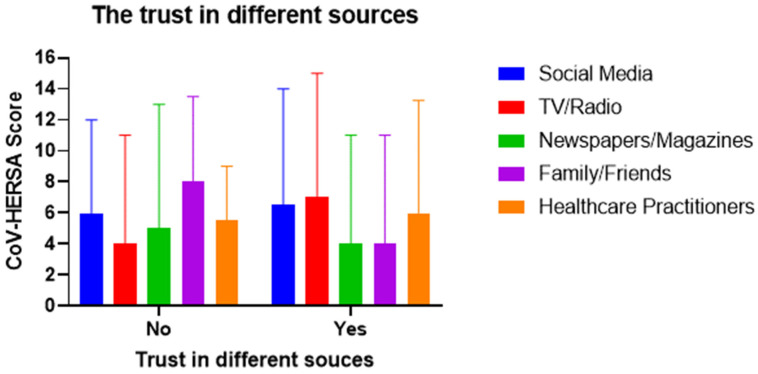
Participants’ CoV-HERSA scores are based on their trust in the different sources of information. The bar chart demonstrates the different CoV-HERSA scores of participants based on whether or not they use the different sources of information. These sources are social media (Blue), TV/radio (Red), Newspapers/magazines (Green), family/friends (Purple), and Healthcare practitioners (Orange). The bar chart demonstrates median scores with interquartile ranges. Two-way ANOVA followed by Tukey correction for multiple comparisons were used.

**Figure 6 tropicalmed-07-00375-f006:**
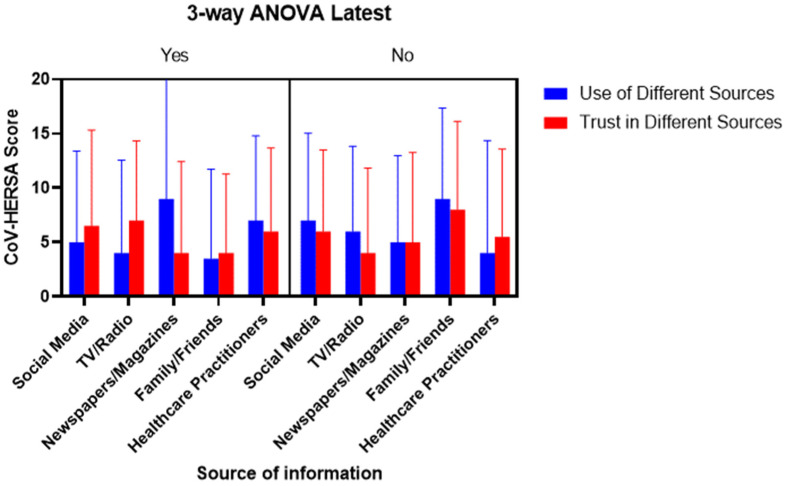
The comparison of CoV-HERSA scores is based on the participants’ use and trust in the different sources of information. The figure is divided into the main section, depending on participants’ use/trust in the different sources of information (Yes) or the lack of them (No). This is followed by comparing bar charts based on participants’ use (Blue) or trust (Red) in the different sources of information. Bar charts demonstrate median CoV-HERSA scores with interquartile ranges. Three-way ANOVA was used here.

**Table 1 tropicalmed-07-00375-t001:** Characteristics of the study participants.

Characteristics	Number (%)
Gender	Male = 193 (49.9%) Female = 194 (50.1%)
Smoking status	Smoker =324 (83.7%) Non-smoker = 63 (16.3%)
Age (years)	34 (24–43) *
Level of education	Middle school 6 (1.6%) High school 85 (22%) Bachelor’s degree 249 (64.3%) Postgraduate degree 47 (12.1%)
Region of residence	Madinah 200 (51.7%) Makkah 94 (24.3%) Riyadh 30 (7.8%) Eastern Province 30 (7.8%) Qassim 29 (7.5%) Al-Baha 1 (0.3%) Jazan 1 (0.3%) Northern borders 2 (0.5%)
Nature of workplace	Educational establishment (student/staff) 192 (49.6%) Healthcare facility (Physician/nurse) 39 (9.8%) Nonmedical establishment (admin/clerk) 60 (15.5%) Field work (Engineer/technician) 21 (5.4%) Home (Housewife/retiree/jobseeker) 76 (19.6%)

* Age is described in median years (interquartile range).

**Table 2 tropicalmed-07-00375-t002:** Participants’ medical experience could influence their CoV-HERSA score.

Characteristics	Number (%)
Personal Medical History	Type I Diabetes 16 (4.1%) Type II Diabetes 15 (3.9%) Cardiovascular diseases 37 (9.6%) Immunodeficiency disorders 2 (0.5%) Autoimmune disorders 6 (1.6%) Malignancy 3 (0.8%) Obesity 74 (19.1%) Healthy 275 (71.1%)
Immediate Family/Friends’ Medical History	Type I Diabetes 157 (40.6%) Type II Diabetes 116 (30%) Cardiovascular diseases 163 (42.1%) Immunodeficiency disorders 9 (2.3%) Autoimmune disorders 46 (11.9%) Malignancy 46 (11.9%) Obesity 127 (32.8%) Healthy 96 (24.8%)
Pregnancy	6 (3.1%)
Pregnant Immediate Family/Friends	Yes = 130 (33.6%) No = 257 (66.4%)
Previous Vaccination (Hep B/Influenza)	Yes = 244 (63%) No = 143 (37%)

Hep B: Hepatitis B.

**Table 3 tropicalmed-07-00375-t003:** Participants’ Experience with COVID-19 Infection.

Characteristics	Number (%)
Diagnosed with COVID-19 via a PCR test	Yes 42 (10.9%) No 345 (89.1%)
Immediate family/friends diagnosed with COVID-19 via a PCR test	Yes 295 (76.2%) No 92 (23.8%)
Hospital admission due to COVID-19	Yes 2 (4.8%) No 40 (95.2)
Hospital admission of immediate family/friends due to COVID-19	Yes 91 (30.8%) No 204 (69.2%)
Death of immediate family/friends due to COVID-19	Yes 120 (40.7%) No 175 (59.3%)

## Data Availability

Raw data will be available as a supplementary upon peer-review completion.

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
