# Peer review of "Assessing the Impact of Use and Trust in Different Sources of Information on COVID-19 Vaccination Uptake in Saudi Arabia (SA) Using the COVID-19 Vaccine Hesitancy and Resistance in SA (CoV-HERSA) Tool"

_tropicalmed, 2022, doi:10.3390/tropicalmed7110375_

Round 1
Reviewer 1 Report
The manuscript by Anwar A. Sayed did a cross-sectional study aiming to dissect the factors associated with SARS-CoV-2 vaccine hesitancy. This is a very interesting study and would be important to let people learn about factors contributing to vaccine hesitancy. Also, the questions and scoring criteria in the survey were well designed. However, the way that the data was presented could be improved and I have some concerns with the statistics as listed below.
Major concerns:
1, The participants of the survey were between 24-43 years old and extremely high rate of them (76.4%) had bachelor’s or higher degree. This population seemed to be selected and not very representative of the whole population that had access to SARS-CoV-2 vaccines.
2, As sample size of different groups varied considerably, presenting individual values in the bar graphs as you did in figure 1 would be very helpful to let the reader know about the sample sizes and variances.
3, The data from the scoring system are discrete, not continuous data, so the data here are nonparametric even if they pass the normal distribution test (Shapiro Wilk Test).
4, The figures have different datasets, so the statistics methods may be different. Please reveal individual statistics method in the figure legend.
5, In figure 5 and figure 6, the variables are not independent of each other, so 2-way ANOVA and 3-way ANOVA are inappropriate.
Minor concerns:
1, Line 41 “both doses of vaccine” →”double doses of vaccine”.
2, Line 78 “person with 30 participants” →”30 participants”
3, Line 199 “study” →”studied”.
Reviewer 2 Report
This manuscript is interesting for both medicine and sociology communities and could be published in Tropical Medicine and Infectious Disease. The topic is definitely original and actual. Covid-19 vaccination has been the cornerstone measure to tackle the severe morbidity and mortality of the ongoing global pandemic. However, vaccine hesitancy and resistance were observed in different populations, including Saudi Arabia (SA), yet such hesitancy was not accurately measured, nor were its influencing factors determined. The Covid-19 Vaccine Hesitance and Resistance in SA (CoV-HERSA) tool was developed, validated, and distributed to 387 participants to measure their Covid-19 vaccine hesitancy and its influencing factors accurately. Different chronic conditions affected participants' CoV-HERSA differently, with those having autoimmune disorders having the highest CoV-HERSA scores. Previous exposure to Covid-19 significantly increased the CoV-HERSA scores. The use and trust of the different sources of information had a significant influence on the CoV-HERSA scores. Those who used newspapers and healthcare practitioners as their source of information had the highest CoV-HERSA scores, while those who relied on family/friends had the lowest scores. The CoV-HERSA is a validated tool that accurately reflects participants' willingness and hesitancy to Covid-19 vaccination and can be used to explore the influence of different factors on the participants' attitudes towards the Covid-19 vaccine. The introduction provide very sufficient background. The research methodology is adequate and modern. The results are clearly presented. The amount of data is large. The conclusions supported by the data. The manuscript good illustrated and interesting to read. English language and style are fine, and may be very minor polishing from native speaker is recommended. I have also couple of minor suggestions:
- The manuscript is quite long, and may be some text could be moved in supplementary information?
- Some more detailed perspectives regarding the future research could be formulated in conclusions section.
Overall, this nice manuscript could be accepted for publication after minor revisions.
Reviewer 3 Report
Due to the limitations which the authors themselves write about, the study can be treated as preliminary to future analyzes. It is known that there are people in every population who do not want to be vaccinated against COVID. Various factors influence participants' attitudes towards Covid-19.
Some factors are common to different populations, while others may be specific to a given region of the world. Therefore, I believe that the manuscript should be published. The results of these and other similar studies are important from the point of view of epidemiology and public health.
Round 2
Reviewer 1 Report
Thank you for the point-to-point responses. As of now, the manuscript is in a obviously better form. However, the issues with statistics persist. I honestly suggest you to seek advice from a statistics technician.
